# Gene Therapy Strategies for Hepatocellular Carcinoma (HCC): Current Landscape and Future Directions

**DOI:** 10.3390/cancers17223608

**Published:** 2025-11-08

**Authors:** Ali Gawi Ermi, Rabha M. Younis, Kayla Rodriguez, Devanand Sarkar

**Affiliations:** 1Department of Cellular, Molecular and Genetic Medicine, Virginia Commonwealth University, Richmond, VA 23298, USA; ali.gawiermi@vcuhealth.org (A.G.E.); rabha.younis@vcuhealth.org (R.M.Y.); 2Massey Comprehensive Cancer Center, Virginia Commonwealth University, Richmond, VA 23298, USA; rodriguezka2@vcu.edu; 3Department of Cellular, Molecular and Genetic Medicine, Massey Comprehensive Cancer Center, Virginia Commonwealth University, Richmond, VA 23298, USA

**Keywords:** HCC, gene therapy, viruses, nanoparticles, clinical trials, CAR-T, CRISPR/Cas9

## Abstract

Hepatocytes are the main cells in the liver, constituting ~90% of all liver cells. The cancer that arises in hepatocytes is known as hepatocellular carcinoma (HCC), accounting for ~80% of all liver cancers. If diagnosed early, HCC can be treated with liver transplantation, tumor resection or local radio- or chemotherapy, providing an overall survival of more than 5 years. HCC patients present with vague symptoms, and the disease is often not diagnosed early. Most HCC patients are diagnosed at an advanced stage, and they are treated with either immunotherapy or a specific class of drugs known as tyrosine kinase inhibitors (TKIs), such as sorafenib. These treatment modalities provide less than 2 years of survival. Additionally, HCC develops as a consequence of chronic liver diseases, such as viral hepatitis, fatty liver disease, or alcoholism, which significantly damage the liver. The damaged liver cannot properly metabolize the anticancer drug, causing severe drug-induced toxicity. In this scenario, alternative treatment approaches are increasingly being sought after, and one modality of treatment is gene therapy. In gene therapy, defective genes are either replaced or corrected, or harmful genes are inactivated. Here, we describe the current status of gene therapy in HCC, its promises, success stories, hurdles, and challenges.

## 1. Introduction

### 1.1. Hepatocellular Carcinoma (HCC): Epidemiology, Etiology, and Clinical Challenges

Hepatocellular Carcinoma (HCC) is the most common type of liver cancer, originating in hepatocytes, the primary cells of the liver [1]. Globally, liver cancer ranks as the sixth most prevalent cancer and the third leading cause of cancer-related deaths [2]. HCC develops primarily in individuals with underlying liver disease, with chronic infections from hepatitis B virus (HBV) or hepatitis C virus (HCV) being major risk factors, as they can cause persistent liver inflammation, leading to cirrhosis, a disease in which healthy liver tissue is replaced by scar tissue, impairing blood flow and liver function [1,3]. Other significant risk factors include heavy alcohol consumption, which can lead to cirrhosis, and metabolic dysfunction-associated steatohepatitis (MASH), a severe form of metabolic dysfunction-associated fatty liver disease (MAFLD) that triggers inflammation, liver cell damage, and cirrhosis [1]. Cigarette smoking and eating foods containing aflatoxin B1, a poison found in fungus, increase HCC risk. Additionally, genetic conditions such as hereditary hemochromatosis, tyrosinemia, alpha-1 antitrypsin deficiency, glycogen storage diseases, porphyria cutanea tarda, and Wilson disease may increase the risk of liver cancer [1]. Figure 1 summarizes major global risk factors that contribute to HCC development.

Despite advancements in HCC treatment, current modalities have significant limitations [4]. Surgical resection and liver transplantation are only viable for patients with early-stage disease and sufficient liver function, excluding many individuals with cirrhosis or multifocal tumors [5]. Moreover, surgical resection is limited by high recurrence rates, exceeding 50%, along with the complexity of liver anatomy, which increases surgical risks and morbidity. Liver transplantation, while an effective curative option, is severely limited by donor organ scarcity, which can lead to tumor progression during the waiting period, potentially rendering patients ineligible for transplant [6]. Non-surgical treatments such as ablation and embolization offer alternatives for localized disease but are limited by tumor size and location, and recurrence remains a concern for ablation [7]. Additionally, radiation therapy, including stereotactic body radiation therapy (SBRT), faces limitations such as toxicity to surrounding healthy liver tissue and a lack of long-term outcome evidence, making it difficult to evaluate its long-term effectiveness [8]. These limitations highlight the need for ongoing research and the development of therapeutic strategies that can overcome the challenges posed by current treatment modalities for HCC.

### 1.2. Rationale for Gene Therapy in HCC

Given the therapeutic limitations of HCC, gene therapy has emerged as a promising strategy to address the gaps in conventional therapies. Gene therapy is an emerging therapeutic strategy involving the use of vectors to deliver exogenous genetic material into target cells, with the goal of modifying gene expression [9]. This can be achieved by replacing defective or absent genes with functional copies or by silencing the expression of pathogenic genes, ultimately aiming to correct or alleviate disease phenotypes. In 1990, gene therapy was first applied clinically in a trial treating two children with Adenosine-deficient Severe Combined Immunodeficiency (ADA-SCID), using retroviral vectors to introduce the *ADA* gene into their T cells [10]. This study marked a pivotal step in demonstrating the feasibility of gene therapy as a therapeutic approach. Since that milestone, gene therapy has advanced significantly, with multiple FDA approvals and expanding applications across various diseases. In 2017, Tisagenlecleucel (trade name: Kymriah) became the first gene therapy approved by the FDA for the treatment of acute lymphoblastic leukemia and diffuse large B-cell lymphoma (DLBCL) [11]. This breakthrough was soon followed by Acixabtagene ciloleucel (trade name: Yescarta), approved later that year for DLBCL and Non-Hodgkin’s lymphoma [12].

One of the primary advantages of gene therapy in the treatment of HCC is its capacity for specificity and targeted delivery. The liver is a prime target for gene therapy due to its essential biological functions, vascular access to the major target cell (hepatocytes), and immunotolerant environment [13]. The liver’s unique vasculature, coupled with the liver’s fenestrated endothelium, allows for efficient access of vector particles directly to hepatocytes. Vectors such as adeno-associated viruses (AAVs) are the leading delivery system for liver-directed gene therapy due to their hepatotropic nature, ability to efficiently transduce hepatocytes, relatively low immunogenicity, and capacity to maintain long-term transgene expression [14]. Additionally, the use of hepatocyte-specific promoters can further restrict gene expression to liver cells, reducing off-target effects and enhancing safety. Patients with autoimmune disease are not suitable for immunotherapy, the current first-line treatment for HCC [15]. Gene therapy, especially using non-viral vectors, might be appropriate for HCC patients with autoimmune diseases [16].

Gene therapy for HCC holds significant promise for personalized medicine, offering tailored treatment strategies that account for the unique genetic and molecular profiles of individual patients. HCC is characterized by a highly complex and heterogeneous genetic landscape, involving numerous somatic mutations and epigenetic alterations that vary significantly among patients [17]. The genetic heterogeneity poses a challenge for generalized “one-size-fits-all” therapeutic strategies. Personalized gene therapy can leverage detailed genomic profiling of a patient’s tumor to identify specific actionable mutations, overexpressed oncogenes, or silenced tumor suppressor genes. This precision can significantly enhance therapeutic efficacy while minimizing off-target effects, directly addressing the complex genetic landscape of HCC.

### 1.3. Scope and Objectives of the Review

This review aims to provide a comprehensive overview of the current landscape of gene therapy approaches for HCC. The following sections will detail gene therapy strategies targeting tumor cells directly, including Suicide Gene Therapy, Tumor Suppressor Gene Replacement, Oncolytic Virotherapy, and Gene Editing techniques like CRISPR/Cas9, as well as strategies that target the Tumor Microenvironment (TME), focusing on Immunogene Therapy, Anti-angiogenic Gene Therapy, and Gene Therapy Targeting Hepatic Stellate Cells (HSCs). Furthermore, various vector design and delivery systems, including both Viral and Non-Viral Vectors and Targeted Delivery Strategies, will be detailed. Finally, this review will summarize key findings from completed and ongoing clinical trials, address challenges and limitations, and discuss future directions and perspectives for gene therapy. Beyond the scope of recent publications, this review presents an extensive update, integrating advanced gene therapy strategies that not only target tumor cells but also modulate the TME. Furthermore, it provides an in-depth analysis of cutting-edge vector design and targeted delivery systems. Additionally, this review will summarize key findings from completed and ongoing clinical trials, address challenges and limitations, and discuss future directions and perspectives for gene therapy.

## 2. Gene Therapy Strategies Targeting Tumor Cells Directly

Gene therapy provides a molecular-level approach to tackling the complex genetic makeup of HCC. Unlike traditional treatments, gene-based interventions can be specifically designed to address the unique genetic drivers in an individual patient’s tumor. HCC is known for its high degree of genetic variability and frequently contains mutations in genes such as *TP53* (tumor protein p53) and *CTNNB1* (catenin beta 1), and the promoter region of *TERT* (telomerase reverse transcriptase), which are often considered difficult to target with conventional drugs [18]. Gene therapy can help overcome these challenges by inserting tumor-suppressor genes or disabling oncogenic sequences directly within cancer cells [19]. This targeted delivery is central to the concept of personalized treatment due to the molecular diversity seen in HCC tumors. Customizing gene therapies based on each patient’s specific genetic profile holds significant promise. This part of the review spotlights therapeutic approaches that utilize gene-based techniques to specifically attack cancer cells at the molecular level. It covers both gene therapy methods that work directly on tumor cells and those that target the cellular environment around tumors. Figure 2 illustrates some of the strategies that target tumor cells directly.

### 2.1. Suicide Gene Therapy Triggering Selective Tumor Cell Death

Suicide gene therapy, also known as gene-directed enzyme prodrug therapy (GDEPT), is a strategy where a gene encoding a drug-activating enzyme is delivered into tumor cells [20]. Once inside, these genes convert a harmless prodrug into a cytotoxic compound, killing the cancer cells. A well-established system is the herpes simplex virus thymidine kinase (HSV-TK) gene in combination with the prodrug ganciclovir (GCV) [21,22]. HSV-TK phosphorylates GCV into a toxic nucleotide analog, GCV-triphosphate, that interferes with DNA synthesis, leading to the death of proliferating tumor cells. Similarly, the cytosine deaminase (CD) gene, typically derived from bacterial sources, converts the prodrug 5-fluorocytosine (5-FC) by deamination into the chemotherapeutic agent 5-fluorouracil (5-FU), which inhibits thymidylate synthase (TS) and is toxic to cancer cells [23]. These enzyme/prodrug systems lead to a bystander effect, where not only transduced cancer cells die, but neighboring tumor cells are also killed due to the transfer of toxic metabolites [20]. To maximize tumor specificity, limit off-tumor expression, and reduce damage to healthy cells, suicide genes can be placed under tumor-selective promoters, e.g., survivin/BIRC5 (baculoviral IAP repeat containing 5) or hTERT promoters active mainly in HCC and other cancer cells [24,25]. It has been particularly useful against solid tumors and chemo-resistant HCC and can even enhance the effectiveness of radiotherapy when combined [26,27,28].

#### 2.1.1. Clinical Application and Trial Outcomes of Suicide Gene Therapy

Early-phase clinical trials have highlighted the feasibility and safety of suicide gene therapy in patients with HCC. In a Phase I study (NCT00844623), an adenoviral vector encoding the HSV-TK gene was directly injected into tumors, followed by systemic administration of ganciclovir (GCV) [29]. This approach demonstrated a favorable safety profile, with most adverse effects being mild. Preliminary evidence of anti-tumor activity, particularly in patients receiving higher viral doses, was noted, with tumor necrosis and metabolic inactivation observed in PET imaging [29]. Building on these findings, a Phase II/III randomized study (NCT00300521) evaluated the same gene therapy vector as an adjuvant treatment following liver transplantation in patients with advanced HCC [30]. The trial reported significantly improved three-year overall survival in the gene therapy group (~69.6%) compared to the transplant-only control group (~19.9%) [30]. In patients without vascular invasion, the survival benefit was even more remarkable, with 100% overall survival and 83.3% recurrence-free survival at three years [30]. Despite these encouraging results, not all studies have yielded consistent outcomes, with some trials discontinued prematurely due to suboptimal gene delivery or insufficient therapeutic response [31]. Nonetheless, most clinical data affirm the tolerability of this strategy, and efforts are now focused on enhancing its efficacy through multimodal approaches.

#### 2.1.2. Enhancing Efficacy of Suicide Gene Therapy with Combination Therapies

To overcome existing limitations and boost treatment outcomes, suicide gene therapy is increasingly being explored in conjunction with other modalities. One promising strategy involves co-delivering immunostimulatory molecules, such as chemokines or cytokines, alongside the suicide gene. For instance, in preclinical HCC models, an adenovirus engineered to express both HSV-TK and a membrane-bound chemokine gene fusion [monocyte chemoattractant protein-1 (MCP-1) fused with the membrane-spanning domain of C-X3-C motif chemokine ligand 1 (CX3CL1)] significantly enhanced immune cell infiltration into the tumor, thereby amplifying the cytotoxic effects of GCV treatment [32]. An adenovirus was created simultaneously expressing a trans-splicing ribozyme targeting hTERT and HSV-TK, along with a miR-122a recognition site, with the hypothesis that high miR-122a levels will lead to degradation of the ribozyme in normal hepatocytes, while low miR-122a levels in HCC cells will allow functioning of the ribozyme, causing degradation of hTERT, expression of HSV-TK, and cell death [33]. This approach has been successfully tested in pre-clinical models [33] and is being evaluated in a Phase 1/2a clinical trial along with valganciclovir in HCC patients (NCT05595473). Research is also progressing toward novel delivery methods, such as ultrasound-triggered microbubbles, nanoparticles (NPs), or integrin-targeted cationic microbubbles, to increase transduction efficiency and therapeutic gene expression. CD/TK double suicide gene was incorporated into cationic microbubbles coated with anti-αVβ3 integrin monoclonal antibody to improve targeted delivery [34]. Treatment with ultrasound, which sheared the microbubbles, along with 5-FC/GCV, resulted in significant suppression of the growth of HepG2 xenografts [34]. Purine nucleoside phosphorylase (PNP) is an E. Coli enzyme that converts the pro-drug fludarabine phosphate (FP) into a cytotoxic compound 2-fluoroadenine-triphosphate (2-FATP), an ATP analog [35]. During DNA and RNA syntheses, incorporation of 2-FATP inhibits these processes, inducing cell death. Human PNP does not metabolize FP. As such, this system works only in those cells that express E. Coli PNP and has been shown to be superior to the HSV-TK system in in vitro studies [36]. The PNP gene, delivered via ultrasonic nanobubbles, induced apoptosis in HepG2 and SMMC7721 cells upon treatment with FP [37]. Intra-tumoral injection of an adenovirus delivering PNP and intravenous (IV) administration of FP have been shown to be safe and efficacious in a Phase I trial for solid cancers, but it is yet to be tested in HCC patients [38]. While combination therapies hold significant promise, persistent obstacles such as incomplete gene transduction, insufficient prodrug activation, and transient gene expression continue to hinder full clinical translation. Clinically, gene therapy vectors have also been paired with locoregional treatments like transarterial chemoembolization (TACE) or radiofrequency ablation (RFA) to improve local tumor control. However, the combination of suicide gene therapy with TACE or RFA is yet to be evaluated.

### 2.2. Restoring Tumor Suppressor Function: Focus on p53

HCC frequently exhibits alterations in tumor suppressor genes, with *TP53* mutations observed in an estimated 50% of cases [39]. Therapeutic strategies focused on compensating for defective or absent tumor suppressor genes have become a cornerstone of cancer treatment research, with the primary goal of reestablishing cellular growth regulation and promoting programmed cell death in malignant cells. Of these approaches, the restoration of p53 function has received the greatest focus in clinical investigations. The p53 protein, encoded by the *TP53* gene, serves as a critical cellular guardian that responds to DNA damage and cellular stress by either halting cell division to allow for repair or triggering apoptosis when damage is irreparable [40]. Given that TP53 mutations are found in approximately half of all human cancers, making it one of the most frequently altered genes in malignancy, the therapeutic potential of restoring p53 function represents a particularly promising avenue for cancer treatment. Several clinical trials have explored the therapeutic potential of introducing wild-type TP53 into tumor cells using adenoviral vectors. Indeed, Gendicine, developed by Shenzhen SiBiono GenTech, represents a recombinant adenovirus expressing TP53 (rAd-p53) and is the first gene therapy product approved for clinical use by the Chinese State Food and Drug Administration [41]. In a clinical study with 40 HCC patients, the efficacy of fractionated stereotactic radiotherapy (fSRT) was evaluated with or without rAd-p53 [42]. The combination was safe and showed better efficacy compared to fSRT alone, with 1-year disease-free survival rates of 90% and 70%, respectively [42]. In advanced HCC, combining rAd-p53 with TACE has shown encouraging results [43]. The trial reported improved survival outcomes in patients receiving both p53 gene therapy and TACE, compared to those treated with TACE alone, suggesting a synergistic effect [43]. Nevertheless, tumor heterogeneity can lead to variable responses; some tumor cells may respond to restored p53 function, while others with alternate driver mutations may not. Tumors may also develop resistance mechanisms, such as upregulation of anti-apoptotic proteins. From a safety standpoint, p53-based gene therapy has generally shown good tolerability. Wild-type p53 is a natural tumor suppressor and is unlikely to harm non-malignant cells. However, concerns remain about potential immune reactions or off-target effects, especially when administered at high doses [9]. Looking ahead, combinatorial strategies may offer the best chance for success.

Beyond p53, other tumor suppressors are also being investigated. The retinoblastoma (*RB1*) gene, frequently inactivated in HCC [44], represents another candidate for gene replacement therapy, although clinical data in this area remain limited. Additionally, researchers have explored the delivery of pro-apoptotic genes that mimic tumor suppressor functions. One notable example is the TRAIL (TNF-related apoptosis-inducing ligand) gene. In a preclinical model, an AAV vector carrying TRAIL under the control of a telomerase promoter (hTERT) achieved tumor-specific expression, effectively inducing apoptosis in HCC cells while sparing normal liver tissue [45]. YAP1 is a downstream effector of the Hippo signaling pathway, which regulates development, growth, repair, and homeostasis. YAP1 binds with TEAD (Transcriptional enhancer-associated domain) to form a functional transcription factor complex, and Vestigial-like protein 4 (VGLL4) is a tumor suppressor protein that binds to YAP1 and prevents formation of the YAP-TEAD complex [46]. An oncolytic adenovirus expressing VGLL4 under the survivin promoter induced cell cycle arrest and apoptosis in HCC cells and inhibited the growth of HuH-7 xenografts in nude mice [47]. Signal transducer and activator of transcription 3 (STAT3) is a potent oncogene that is negatively regulated by suppressor of cytokine signaling 1 (SOCS1). An oncolytic adenovirus expressing SOCS1 inhibited STAT3 and its downstream molecules, notably survivin, cyclin D1, Bcl-xL and c-Myc, and induced apoptosis in HCC cells [48]. Potassium voltage-gated Channel subfamily Q member 1 (KCNQ1) is a tumor suppressor that is hypermethylated in human HCC, and KCNQ1 interacts with β-catenin inhibiting its activity [49]. Overexpression of KCNQ1 inhibited in vitro invasion and in vivo lung metastasis by Hep3B cells [49]. Vacuolar protein sorting 4 homolog a (VPS4A) regulates exosome-mediated miRNAs, which play an important role in tumorigenesis. Overexpression of VPS4A interfered with miRNAs regulating phosphatidylinositol-3-kinase/Akt (PI3K/Akt) pathway, thereby inhibiting this pathway and inhibiting xenografts of human HCC cells, such as Hep3B, in nude mice [50]. Other examples of tumor suppressor genes inhibiting in vitro and in vivo growth of HCC cells include catenin alpha 3 (CTNNA3), CCAAT/enhancer-binding protein alpha (CEBPA), and protocadherin 9 (PCDH9) [51,52,53]. These are candidate molecules that need to be further explored to evaluate the potential therapeutic efficacy in clinical settings. Despite these promising developments, gene replacement therapy in HCC faces considerable challenges. The genetic heterogeneity of tumors often limits the effectiveness of single-gene correction approaches, especially in tumors harboring multiple oncogenic drivers. Therefore, restoring a single tumor suppressor may not be sufficient as a standalone treatment.

### 2.3. Oncolytic Virotherapy (OV): Harnessing Viruses for Targeted Tumor Destruction

OV is a form of cancer treatment that involves the use of viruses specifically engineered or selected to target and destroy tumor cells [54]. These viruses are designed to replicate exclusively within cancerous cells, leaving normal tissue largely unaffected. As the virus multiplies inside the tumor cell, it eventually causes the cell to rupture [55,56]. This not only eliminates the infected cell but also helps activate the immune system to recognize and attack the remaining cancer cells [55].

#### 2.3.1. Genetically Engineered Viruses for Tumor Selectivity

The use of genetically modified oncolytic viruses marks a significant advancement in cancer therapy, drawing on extensive progress in molecular biology and virology. Through precise genetic alterations, these viruses are tailored to selectively target tumor cells, enhance treatment outcomes, and minimize harm to healthy tissue [57]. One important modification strategy involves altering viral genes that interact with the host’s interferon signaling pathways, allowing the virus to replicate more effectively within cancer cells that often have defective antiviral responses [58]. This targeted replication reduces the risk of off-target effects in normal tissues. Another powerful engineering technique involves the integration of cancer-specific promoters, which regulate genes necessary for viral replication selectively in target cancer cells [59]. In HCC, the Glypican 3 (GPC3) promoter can be employed to ensure that viral replication is largely confined to malignant liver cells, thereby limiting unwanted viral activity in non-cancerous organs [60,61]. Furthermore, advancements in bioengineering, such as ultrasound nano- or microbubbles, have facilitated delivery of oncolytic viruses and therapeutic genes directly to the tumor site [62]. OVs such as the modified vaccinia virus strain JX-594 (Pexa-Vec) have been developed to express granulocyte-macrophage colony-stimulating factor (GM-CSF), providing a dual therapeutic mechanism through direct tumor cell destruction and stimulation of the immune system [63,64]. Early-phase clinical studies have indicated that such treatments exhibit both immunostimulatory activity and a favorable safety profile [63,64].

#### 2.3.2. Mechanisms of Oncolysis and Immune Activation

The two main mechanisms by which OVs exert their effects are direct killing of cancer cells and activation of the immune system [65,66]. The first effect happens when the virus enters a tumor cell and uses the cell’s machinery to reproduce. This disrupts normal cell function and eventually leads to cell death. Depending on the virus and the cell type, this may occur through processes like apoptosis (programmed cell death), necrosis (cell injury), or autophagy (self-digestion). The second, and often more impactful, effect involves the immune system. When infected cancer cells die, they release viral components and tumor-related proteins into the surrounding area. These molecules alert the immune system and trigger what is called immunogenic cell death, a process that transforms the tumor into an environment that actively attracts immune cells [67,68]. The immune response starts with the innate immune system, which acts as the body’s first line of defense [69]. Viral infection stimulates receptors that detect foreign material, prompting the recruitment of cells like neutrophils, macrophages, and dendritic cells. These cells not only help clear infected cells, but also present tumor antigens to other immune cells. This leads to the activation of the adaptive immune system, which provides a longer-lasting defense [70]. Dendritic cells present antigens to T cells, leading to the production of tumor-specific cytotoxic T cells. These T cells can then target and kill cancer cells, including those at distant sites. In some cases, this immune memory may help prevent the cancer from returning. By combining direct tumor killing with long-term immune activation, oncolytic viruses offer a powerful, dual-pronged approach that can produce results greater than either effect alone.

#### 2.3.3. Clinical Evidence and Safety Considerations

Bringing OVs from laboratory experiments to real-world treatment for HCC has shown encouraging progress. Several virus-based therapies have entered clinical trials for HCC, and early results suggest that they can be both effective and safe when used in patients. One of the most studied therapies in this area is Pexastimogene devacirepvec, also known as JX-594 (Pexa-Vec). It is a modified vaccinia virus (Wyeth strain) with a deletion in the viral thymidine kinase gene and the GM-CSF gene to stimulate immune responses. A phase 1 clinical trial [NCT00629759] involving 14 patients with primary or metastatic liver cancer established the maximum tolerated dose (MTD), leading to multiple phase 2 clinical trials. These trials include sorafenib naïve advanced HCC patients [NCT01636284], advanced HCC patients who failed sorafenib treatment [NCT01387555], and patients with unresectable primary HCC [NCT01171651, NCT00554372]. Collectively, these clinical trials have shown that patients with advanced HCC who received higher doses of JX-594 experienced better survival and measurable tumor shrinkage. Side effects were generally mild and temporary, with flu-like symptoms being the most common. Interestingly, patients whose tumors had certain genetic features—such as changes in the EGFR/Ras pathway—seemed to respond exceptionally well, highlighting the growing importance of matching treatments to a patient’s tumor profile [71]. PHOCUS is a Phase 3, randomized, open-label study of sequential therapy with Pexa-Vec (JX-594) and sorafenib in patients with advanced HCC [72]. A total of 459 patients, from 142 sites in 16 countries, were randomly assigned to 2 treatment arms: 234 patients in the Pexa-Vec followed by sorafenib treatment group and 225 patients in the sorafenib only treatment group [72]. At the interim analysis, the median overall survival (OS) was 12.7 months (95% CI: 9.89, 14.95) in the Pexa-vec plus sorafenib arm and 14.0 months (95% CI: 11.01, 18.00) in the sorafenib arm, resulting in the early termination of the study [72]. Although the findings of this trial are disappointing, the potential application of Pexa-vec with other agents, especially immunotherapy, cannot yet be ruled out. Another promising candidate is VG161, an oncolytic herpes simplex virus that has been engineered to deliver multiple immune-boosting molecules, such as IL-12, IL-15, IL-15Rα, and a PD-1–PD-L1-blocking fusion protein [73]. In a Phase 1 trial with 44 liver cancer patients, including 40 HCC patients, VG161 showed a good safety profile, with seven patients showing a partial response and 17 maintaining stable disease [73]. The median progression-free survival (PFS) was 2.9 months (95% confidence interval, 1.81–3.70), and the overall survival (OS) was 12.4 months (95% confidence interval, 7.10–20.10) [73]. Notably, patients who were previously treated with checkpoint immunotherapy (CPI) for more than 3 months showed extended PFS and OS, compared to those who were on CPI for less than 3 months [73]. The built-in immune checkpoint inhibitors in VG161 represent a creative way to combine two types of immunotherapies in one platform. These encouraging findings pave the way for further testing of VG161 in expanded and stringent Phase 2/3 trials. H101 (Oncorine) is a human recombinant type 5 adenovirus, in which the gene *E1B*, coding the anti-apoptotic E1B55K protein that deactivates p53, and a portion of the E3 region have been deleted. Originally named ONYX-015, the mechanism of tumor-specific replication of H101 was shown to be late viral RNA export, rather than p53 inactivation [74]. In a Phase I trial, 87 HCC patients received transarterial injection of H101 and TACE while 88 patients were treated with TACE alone [75]. For H101+TACE and TACE only groups, respectively, the following parameters were observed: complete response (CR): 28.7% and 14/8%, partial response (PR): 32.2% and 21.6%, stable disease (SD): 26/4% and 38.6% and progressive disease (PD): 12.6% and 25%, OS: 12.8 and 11.6 months, and PFS: 10.49 and 9.72 months [75]. In a pilot study, 18 patients, who failed prior systemic therapy, were treated with H101 and nivolumab. The combination treatment was well-tolerated with no grade ¾ events, and median PFS and OS were 2.69 and 15.04 months, respectively [76]. Ongoing research is focused on improving patient selection, finding the right dosing strategies, and exploring how these viruses can be combined with other treatments to boost effectiveness without increasing risks.

### 2.4. Clustered Regularly Interspaced Short Palindromic Repeats/CRISPR-Associated Protein 9 (CRISPR/Cas9) Gene Editing in HCC: Precision Medicine at the Genomic Level

The genetic nature of HCC makes it suitable for direct treatment using gene-editing tools such as CRISPR/Cas9. In recent years, the CRISPR/Cas9 system has gained widespread application in HCC research due to its ability to target virtually any gene and its relatively short experimental timelines [77,78]. It has been utilized for various purposes, including the screening of oncogenes and tumor suppressor genes, the generation of HCC animal models, and the identification of novel biomarkers.

This approach allows for the development of personalized therapies that may provide better treatment outcomes than conventional methods. The CRISPR/Cas9 system, identified initially as part of the adaptive immune defense in prokaryotes [79], has since been developed into a powerful and widely adopted gene-editing technology through extensive research and bioengineering [80,81,82]. CRISPR/Cas9 functions through a coordinated mechanism involving three key components: the Cas9 endonuclease, which introduces precise double-stranded breaks in DNA, a single guide RNA (sgRNA) that determines the target sequence by complementary base pairing, and the genomic DNA site designated for editing [82]. This sequence-specific interaction supports its potential use as a personalized treatment modality within the context of HCC. It offers a high degree of accuracy in targeting and modifying genes associated with cancer initiation and progression. Its application enables researchers to deactivate oncogenic drivers, repair deleterious mutations, and enhance the immune system’s ability to recognize and eliminate malignant cells.

#### 2.4.1. Gene Knockout, Knock-In, and Functional Modulation

CRISPR technology offers multiple strategies for genetic intervention in HCC, primarily categorized into gene knockout, knock-in, and transcriptional modulation. Knockout approaches, the most used, rely on Cas9 nuclease to introduce double-strand breaks (DSBs) at targeted loci. These breaks are typically repaired by non-homologous end joining (NHEJ), a process that often introduces insertions or deletions (indels), effectively disrupting the gene’s function [83].

Knock-in strategies, on the other hand, involve supplying a repair template that enables precise gene correction via homology-directed repair (HDR) [84]. While HDR is more technically demanding and less efficient in vivo, it holds great promise for correcting point mutations in genes such as CTNNB1, TP53, or the TERT promoter, commonly observed in HCC. Emerging techniques like base editing and prime editing offer alternatives that avoid double-strand breaks and enable single-base corrections, potentially increasing the precision and safety of genome editing in cancer therapy [85,86]. CRISPR is also being adapted for non-cutting gene regulation through systems like CRISPR activation (CRISPRa) or CRISPR interference (CRISPRi) [87]. These utilize catalytically inactive Cas9 (dCas9) fused to transcriptional activators or repressors to modulate gene expression without altering DNA sequences. CRISPRa has been used to re-activate silenced tumor suppressor genes, while CRISPRi has helped identify essential oncogenic drivers in HCC cells.

#### 2.4.2. Delivery and Specificity: Major Barriers to Clinical Translation

Despite its promise, the effective and safe delivery of CRISPR-Cas9 components to tumor sites remains a significant challenge in treating HCC. The system’s size and complexity consisting of a large Cas9 protein (~160 kDa) and a guide RNA make direct intracellular delivery difficult. To overcome these limitations, Lipid nanoparticles (LNPs), polymer-based carriers, and ligands targeting hepatic cells offer a non-viral strategy for delivering CRISPR/Cas9 systems, demonstrating potential for precise gene editing in liver tissues and hepatocellular carcinoma, while minimizing off-target activity. For instance, LNPs and polymer-based carriers have shown success in delivering mRNA or Cas9 ribonucleoprotein (RNP) complexes to liver tumors in preclinical models. Some designs utilize ligands for asialoglycoprotein receptors (ASGPR) to direct CRISPR/Cas9 delivery to hepatocytes or HCC cells, achieving tumor-specific editing with minimal off-target effects.

Lactose binds to asialoglycoprotein receptors, expressed by HCC cells. A lactose-derived branched cationic biopolymer (LBP) was used to deliver pCas9-survivin to knockout survivin [88]. The conjugated LBP was administered via the tail vein and showed efficient in vivo gene editing and anti-tumor efficacy against orthotopic xenografts of BEL7402 cells in nude mice [88]. NFE2L2 (NFE2 like bZIP transcription factor 2) is a transcription factor mediating antioxidant response. Sonodynamic therapy (SDT) works by generating reactive oxygen species (ROS) and compensatory increase in NFE2L2 limits the efficacy of SDT [89]. An ultrasound-controlled LNP system was used to deliver sgRNA for NFE2L2, which significantly augmented the efficacy of SDT in a xenograft model of HepG2 cells [89]. In two elegant studies, T cells isolated from HCC patients were modified by CRISPR/Cas9 and the efficacy of these T cells in inhibiting tumor growth was demonstrated in humanized HCC PDX models [90,91]. In the first study, PD-1 was deleted in tumor infiltrating lymphocytes (TILs) [90], while in the second study, sarcoma homology 2 domain-containing protein tyrosine phosphatase-1 (SHP-1) was deleted in CD8^+^ T cells [91]. Glypican-3 (GPC3) is overexpressed in HCC [92]. It is a 70 kDa heparan sulfate proteoglycan attached to the exocytoplasmic surface of the plasma membrane and activates pro-tumorigenic Wnt and Hippo signaling pathways [92]. Ultra large porous silica nano-depot (UPSND) was used to deliver Cas9 and sgRNA for GPC3 in a Hepa1-6 mouse orthotopic allograft model demonstrating efficient inhibition of Wnt and Hippo signaling, complete tumor eradication and T cell infiltration in the tumor [93]. A polyamidoamine-aptamer-coated hollow mesoporous silica NP was used for the co-delivery of sorafenib and CRISPR/Cas9 for epidermal growth factor receptor (EGFR) [94]. This system, delivered IV in mice, showed marked inhibition in growth of tumors generated from H22, murine HCC cells [94]. Similar approaches were used to target survivin [95].

These examples demonstrate that CRISPR/Cas9 approach is showing promise. Nonetheless, efficient delivery to all tumor regions, especially in multifocal or fibrotic livers, remains difficult [96]. Moreover, off target editing remains a critical concern [97]. Cas9 may cut unintended genomic sites with similar sequences to the target, potentially disrupting genes in normal cells or leading to harmful mutations. To address this, researchers are refining guide RNA design, using high-fidelity Cas9 variants, and limiting Cas9 expression through transient delivery strategies [98]. Tumor heterogeneity further complicates CRISPR application. Since HCC tumors often harbor diverse genetic clones, a single guide RNA may not adequately target all cancer-driving mutations. Combinatorial CRISPR strategies, using multiple guide RNAs to target several key pathways simultaneously, have been proposed. Immune responses to the bacterial-origin Cas9 protein are another potential barrier. Immune recognition may limit the duration of Cas9 activity or cause inflammation, thus, transient immunosuppression or “stealth” delivery systems are being explored to reduce immunogenicity [99]. This ex vivo approach circumvents many delivery challenges by modifying immune cells outside the body and reintroducing them to enhance tumor-specific immunity. Early results indicate acceptable safety and feasibility.

## 3. Gene Therapy Strategies Targeting the Tumor Microenvironment (TME)

The TME in HCC arises in the context of chronic liver inflammation, resulting in a dynamic interplay among immune and stromal components [100]. This niche includes various fibroblasts and immune cell populations that modulate tumor development, progression, and metastatic potential. The complexity of this environment contributes to therapeutic resistance and metastatic dissemination, posing significant challenges for treatment strategies. Therefore, TME significantly influences the development, progression, and treatment resistance of HCC, motivating the design of gene therapy approaches that modulate its complex components. Figure 3 shows strategies that target the tumor microenvironment (TME).

### 3.1. Immunogen Therapy

One promising direction involves immunomodulatory gene therapies aimed at enhancing the body’s immune response against tumors by introducing genes encoding cytokines or immune-activating molecules directly into the tumor milieu. For example, adeno-associated viral (AAV) vectors have been engineered to deliver interleukin-12 (IL-12) in a tumor-selective manner using tetracycline-inducible K19 riboswitch regulatory elements that minimize systemic toxicity [101]. These approaches have led to increased interferon-gamma (IFN-γ) levels and infiltration of CD8^+^ T cells, resulting in tumor suppression in HCC models [101].

### 3.2. Anti-Angiogenic Gene Therapy

In addition to modulating immune responses, anti-angiogenic gene therapy strategies seek to impair tumor vasculature while also modulating immune activity. Recent advancements in NP technologies, particularly LNPs, have enabled targeted delivery of siRNAs against angiogenic drivers like vascular endothelial growth factor (VEGF). It was demonstrated that CXCR4-targeted LNPs delivering VEGF siRNA in HCC models suppressed VEGF expression and curbed both angiogenesis and tumor progression [102]. Co-delivery of VEGF siRNA with natural compounds like phenethyl isothiocyanate was shown to further enhance these therapeutic effects [103].

### 3.3. Gene Therapy Targeting Hepatic Stellate Cells (HSCs)

An emerging focus in TME-targeted gene therapy involves hepatic stellate cells (HSCs), which play a key role in liver fibrosis and extracellular matrix remodeling. Leveraging their natural uptake of vitamin A, researchers have designed vitamin A-functionalized NPs to deliver therapies selectively to HSCs. For instance, a study used vitamin A-functionalized fluorinated peptide-lipid hybrid NPs to co-deliver sorafenib and siRNA against heat shock protein 47 (HSP47), leading to marked suppression of fibrotic gene expression and restoration of liver function in mouse models of liver fibrosis [104]. Additional research has validated similar strategies using vitamin A-modified zeolitic imidazolate framework-8 (ZIF-8) NPs for co-delivery of TGF-β1 siRNA and antifibrotic drugs like pirfenidone, demonstrating effective suppression of hepatic fibrosis markers and improvement in liver function [105]. These approaches were tested only for fibrosis, but they have the potential to improve the effectiveness of complementary cancer therapies by overcoming stromal barriers. Liver fibrosis was induced in mice by treatment with CCl_4_ followed by orthotopic implantation of the murine HCC cell HCA-1 [106]. These mice were then treated with lipid/calcium/phosphate/protamine (LCPP) NPs delivering TRAIL plasmid in combination with sorafenib [106]. It was demonstrated that this strategy not only inhibited tumor growth but also attenuated liver fibrosis by reverting activated HSCs to a quiescent state or by inducing apoptosis in the HSCs [106]. Recombinant vesicular stomatitis viruses (rVSVs) are oncolytic viruses that replicate in tumor cells as well as in activated HSCs, thereby killing them, while sparing quiescent HSCs [107]. It was demonstrated that in a rat model of thiocetamide-induced fibrosis, hepatic arterial infusion of rVSV not only induced killing in tumor cells but also caused reversal of progression of fibrosis [107]. In conclusion, gene therapy strategies that modify the TME in HCC—whether by enhancing immune activity, disrupting tumor vasculature, or reversing fibrosis—are emerging as viable adjuncts to conventional treatment. As novel delivery platforms and gene constructs continue to be optimized and tested in early clinical trials, these approaches are likely to become integral components of combination therapies for HCC.

## 4. Chimeric Antigen Receptor (CAR)-T Cells

CAR-T cell-mediated immunotherapy harnesses the power of both gene and cell therapy [108]. In this ex vivo gene therapy, T cells are isolated from a patient and are genetically modified to express a chimeric antigen receptor, which allows the engineered T cells to recognize cancer cells that specifically express the target antigen [109]. The CAR-T cells are reintroduced into the patient, where they target and eliminate the antigen-expressing cancer cells. In general, CARs are composed of three domains, an extracellular antigen recognition domain, a transmembrane domain and an intracellular signaling domain [109,110]. Many clinical trials are ongoing to evaluate CAR-T therapy for HCC for a wide variety of gene products, most notably GPC3. A total of 13 HCC patients were treated with GPC3 CAR-T cells [111]. These patients showed grade 1/2 cytokine release syndrome (CRS) and their OS at 3 years, 1 year, and 6 months were 10.5%, 42.0%, and 50.3%, respectively [111]. CAR-T cells expressing both GPC3 and IL-15, which promote survival of T cells, were evaluated in pediatric and adult HCC patients [112]. While GPC3 CAR-T cells did not show any objective antitumor response, GPC3-IL-15 CAR-T cells induced a 66% disease control rate and 33% anti-tumor response [112]. CT017 CAR-T cells were engineered to express GPC3 and runt-related transcription factor 3 (RUNX3), which facilitates CD8^+^ T-cell infiltration into the TME [113]. In a Phase 1 trial with 7 pre-treated HCC patients, CT017 treatment showed grade 2/3 CRS and achieved 3.5 months median PFS and 7.9 months OS [113]. There are several clinical trials that are ongoing using GPC3 CAR-T cells for HCC, such as, NCT03146234, NCT03884751, NCT03130712, NCT03198546, NCT02715362, NCT04121273, NCT02959151, NCT03084380 and NCT03302403. Mucin-1 (MUC-1), epithelial cell adhesion molecule (EpCAM) and cluster of differentiation 147 (CD147) are expressed on the surface of HCC cells [114,115,116] and CAR-T cells for these two molecules are being evaluated in multiple clinical trials, such as NCT02587689, NCT03013712, NCT02729493 and NCT03993743, the results of which are yet to be determined. Alpha-fetoprotein (AFP), a 70 kDa glycoprotein is an established biomarker for HCC. AFP is a secreted protein and not a suitable target for CAR-T therapy. CAR-T cells directed against an AFP-MHC complex inhibited HCC xenografts of HepG2 cells in vivo [117]. A clinical trial for HCC patients, NCT03349255, using ET1402L1-CAR-T cells, directed against anti-HLA-A02/AFP complex, was terminated and no current trial targeting AFP is ongoing. For solid tumors, there are several limitations, such as cell survival time, negative interactions with TME and issues with antigen affinity, which dampen the efficacy of the CAR-T cells [118]. In addition, side effects, such as off-target effects (off-target/off-tumor toxicity), target-miss effect (on-target/off-tumor toxicity), and cytokine release syndrome (on-target/on-tumor toxicity), also impair efficient use of CAR-T cells [119].

## 5. Vector Design and Delivery Systems

Gene therapy works by introducing genetic material into cellular nuclei, a process mediated by delivery systems known as vectors. Vector design plays a crucial role in the treatment of HCC, with recent advancements focusing on improving cancer cell-targeting, efficacy, and safety. In gene therapy, vector design refers to the engineering and modification of these delivery vehicles, often viruses or NPs, to carry and deliver therapeutic genes into target cells. This process involves selecting the appropriate vector type, optimizing its structure, and tailoring its properties to ensure effective gene delivery, expression, and minimal adverse effects [120]. A variety of vector types are currently employed, as detailed in the following sections.

### 5.1. Viral Vectors

Viral vectors are modified viruses that are used as a tool to deliver genetic material into the cells. The choice of viral vector depends on the specific application, including the target cells, the desired duration of gene expression, the size of the gene to be delivered, and the desired immune response [121]. Viral vectors are at the forefront of gene therapy, which aims to treat genetic diseases by modifying a patient’s genes [121,122]. Their applications in gene therapy include: (a) Gene replacement: delivering a healthy copy of a gene to replace a faulty or missing one (e.g., in cystic fibrosis, hemophilia, certain inherited blindness, spinal muscular atrophy) [123], (b) Gene silencing: delivering genetic material [like short-hairpin RNA (shRNA)] to “turn off” or reduce the expression of a harmful gene or an oncogene [124], (c) Gene addition: introducing new genes to provide a beneficial function (e.g., genes that make immune cells better at fighting cancer) [125], and (d) Gene editing: delivering components of gene-editing tools (like CRISPR/Cas9) to precisely modify DNA within cells [126]. Some of the common viruses used in gene therapy for HCC are described here.

#### 5.1.1. Adeno-Associated Viruses (AAVs)

AAV is derived from the parvovirus and contains a single-stranded approximately 4.7 kb DNA genome [127]. There are several serotypes of AAVs, and each serotype of AAV has a unique approach for infecting host cells [127]. AAV is among the most widely used and effective viral vectors today, with safety and efficacy demonstrated in several human clinical trials. AAVs are known for their low immunogenicity, their ability to infect both dividing and non-dividing cells, and their various serotypes for tissue targeting, making them widely used in gene therapy [128]. Although AAV vectors are effective, there are still concerns about genotoxicity, particularly the risk of HCC development due to integration into specific genomic loci [129,130]. Many factors, such as vector size limited to 4.5 kb, enhancer/promoter selection, and timing of delivery, are critical in mitigating these risks [130]. The natural tropism of AAV serotype 2 vectors for HCC compared to nonmalignant liver cells has been reported in mice and human tissue, resulting in significantly more vector genomes serving as templates for transcription in the cell nucleus [131,132].

#### 5.1.2. Adenoviruses (AdVs)

AdV vectors are another type of viral vector used in gene therapy for HCC, with human serotype 5 adenovirus being the most common. It is a double-stranded DNA virus that can carry larger genetic payloads up to 48 kb and infect a broad range of cell types, both non-dividing and dividing, due to a unique tropism [133]. One limitation of AdV-based gene therapy is its tendency to elicit a stronger immune response, which can be advantageous for some applications (like vaccines) but poses a challenge for sustained gene expression [134,135,136]. In a normal cell following AdV infection, the viral protein E1B inactivates the TP53 signaling pathway, thereby inhibiting TP53-induced apoptosis of these cells, allowing viral replication [137]. For gene therapy using AdV vectors, an essential viral gene (typically E1 and often E3, E2, E4) is deleted to prevent their replication in normal cells [137]. The deleted genes are replaced with a cassette that expresses the therapeutic gene. They are primarily used for gene transfer, delivering a therapeutic gene that can, for example, induce apoptosis, restore tumor suppressor function, or stimulate an immune response [138,139]. In the case of a tumor cell, p53 is usually mutated. Hence, the viral genome replicates, which results in eventual cell death by oncolysis. As described in Section 2.3, oncolytic AdVs are being extensively used for gene therapy of HCC.

#### 5.1.3. Lentiviruses (LVs)

LVs, used in gene therapy, belong to the retroviridae family of viruses and are modified from Human Immunodeficiency Virus-1 (HIV-1) [140]. The LV genome comprises a single-stranded RNA of ~8 kb, which is reverse transcribed into double-stranded DNA during DNA replication, and can integrate its genetic material into the host cell’s genome. This leads to stable, long-term gene expression. LVs can cross the nuclear membrane via nuclear pores and can infect both dividing and non-dividing cells. LVs are used extensively in ex vivo gene therapy, where cells are modified outside the body and then reintroduced to patients, such as in CAR-T therapy. LVs encoding short-hairpin RNA (shRNA) are extensively used to modify cells [141]. However, even though LVs exhibit low immunogenicity, they may also pose a safety risk as they can lead to unnecessary non-target insertion mutations [142]. One potential way to mitigate that effect is to use LVs deficient of integrase, the enzyme required for LV integration [143]. A recent study evaluating 783 patients over more than 2200 total patient-years demonstrated that integration-induced insertional mutagenesis may not be a significant event, thereby establishing the safety of LV-engineered T cell therapy [142].

#### 5.1.4. Herpes Simplex Viruses (HSVs)

HSV is a DNA virus that can carry very large genetic payloads (≥30 kb) and with strong replication capabilities and low toxicity, making it a prominent vector in cancer gene therapy [144]. HSV can infect both dividing and non-dividing cells, but unlike LV, HSV does not integrate into the genome. Oncolytic HSV has shown promising results for HCC patients [144]. However, there are still challenges, such as technological problems of generating high-titer HSVs, pre-existing immunity against HSVs in humans, and a potential for recombination between the wild-type HSV genome and latently infected cells [145].

### 5.2. Non-Viral Vectors

Non-viral vectors, which include inorganic particles, lipids, polymeric NPs, or combinations of different types, are promising gene delivery systems attracting researchers to explore more. They are low in their cytotoxicity, immunogenicity, and mutagenesis, but there are challenges, such as gene transfer efficiency, specificity, gene expression duration, and safety [146]. In comparison to viral vectors, which rely on their natural ability to transfer genes into cells, non-viral gene delivery systems use physical force or the cellular function of endocytosis to facilitate gene transfer to target cells. Moreover, non-viral vectors have demonstrated robust gene loading capacity, high safety and practicability, and simplicity of preparation, particularly for LNPs and cationic polymers [147]. Clinically, 30% of gene therapy clinical trials have been conducted using non-viral vectors [147].

Nanotechnology involves the development and application of materials at the nanoscale range (1 to 300 nm). In nanocarrier delivery systems, NPs are designed to overcome resistance and enhance drug accumulation and sensitivity by delivering agents directly to the tumor cells [148]. NPs are materials at the nano level that have unique physicochemical properties compared with bulk material. Because of these properties, they are able to absorb anticancer agents either on their surface, encapsulated within their core, or both [148]. There are different types of nanocarriers with modifications in their construction or surface properties [149,150]. Here, we provide examples that are used for gene delivery in HCC.

#### 5.2.1. Inorganic Nanoparticles (INs)

INs have gained significant attention as therapeutic candidates due to their tunable size, unique optical and electrical attributes, magnetic, catalytic properties, good biocompatibility, prolonged circulation time, and cellular uptake [151]. These include metals, semiconductors, and metal oxide NPs [151]. In small-animal models, inorganic carriers have shown promising results in the efficacious and safe delivery of nucleic acids to treat oncological diseases [152]. The commonly used INs are composed of noble metals, such as gold (Au)/silver (Ag), iron oxide (Fe_3_O_4_), and silica (SiO_2_). Three metal-based NPs, gold, silver, and platinum, have been studied for their anticancer effect against HepG2 cells, demonstrating effective inhibition of cell viability [153]. Tumor suppressor miR-34a, delivered into HepG2 cells by using ZSM-5 zeolite nanoparticles (ZNP), ZNP/PEI/miR-34a nano-formulation, showed good biocompatibility [154]. Five injections of ZNP/PEI/miR-34a nano-formulation in HCC induced in male BALB/c mice significantly inhibited tumor growth and resulted in a significant decrease in AFP level and liver enzyme activities [154]. Zeolites are widely studied as drug-carrying nanoplatforms to enhance the specificity and efficacy of traditional anticancer drugs [155].

#### 5.2.2. Lipid Nanoparticles (LNPs)

LNPs are spherical structures with a size range from 50 to 200 nm, composed of an amphipathic phospholipid bilayer and an internal aqueous core [156]. Their core–shell nanostructure makes them suitable for loading both hydrophobic and hydrophilic molecules. Normally, hydrophobic drugs are encapsulated in the lipophilic bilayers of the shell, and hydrophilic drugs are entrapped in the aqueous phase of the core. They are biocompatible and naturally biodegradable, which makes them a good candidate as a delivery system [157,158]. Due to their remarkable success as a delivery platform for COVID-19 mRNA vaccines, they gained much attention and are widely used in small-molecule drug and nucleic acid delivery [156,158]. Optimized LNPs with an average particle diameter of ~60 nm exhibited excellent pharmacokinetics and stability following IV administration to HCC-bearing mice and were able to selectively deliver siRNA against the oncogene midkine to HCC with a high degree of efficiency [159]. In combination with sorafenib, this approach effectively eradicated sorafenib-resistant HCC in mice [159]. Lactobionic acid-modified LNPs were developed to co-deliver camptothecin and miR-145, displaying a synergistic efficacy against a diethylnitrosamine/carbon tetrachloride (DEN-CCl_4_)-induced HCC model [160]. ALN-VSP is an LNP composed of siRNAs against kinesin spindle protein (KSP) and VEGF at a 1:1 ratio [161]. A phase 1 clinical trial with advanced HCC patients demonstrated that IV administration of ALN-VSP was well-tolerated and prolonged disease stabilization, especially in patients with extensive metastasis [161]. MTL-CEBPA is an LNP delivering CEBPA small activating RNA, upregulating expression of CEBPA gene [162]. A phase 1 trial, NCI02716012, in advanced HCC patient’s IV administration of MTL-CEBPA was well-tolerated and out of 24 participants, one patient displayed an objective response and 12 patients maintained stable disease [162]. Seven patients received additional TKI treatment after cessation of MTL-CEBPA treatment, out of which three patients showed a complete response, one patient showed partial response, and two patients maintained stable disease [162]. A clinical trial, NCT07053072, is currently ongoing using PD-1 mRNA LNP vaccine for advanced HCC patients. LNPs delivering mRNA for costimulator Oxford 40 ligand (OX40L) showed promising anti-HCC efficacy in a pre-clinical model [163].

#### 5.2.3. Polymeric Nanoparticles (PNPs)

PNPs can be prepared using natural or synthetic polymers and formulated as a reservoir (nanocapsules) or matrix system (nanospheres). To facilitate gene delivery, these polymers can be conjugated with genetic material via electrostatic attraction at physiological pH [164]. The drug can be entrapped within the PNPs by conjugation, adsorption, or encapsulation on the surface of the NP [165]. Various factors affect the gene transfection efficiency of cationic polymers, including their surface charge, structure, and molecular weight. Over the last decade, synthetic polymers that have emerged for use in gene therapy include poly(l-lysine), poly(l-ornithine), linear and branched polyethyleneimine, diethylaminoethyl-dextran, poly(amidoamine) dendrimers, and poly (dimethylaminoethyl methacrylate) [164,165]. Administration of biodegradable poly(beta-amino ester) (PBAE) NPs to a subcutaneous HCC mouse model confirmed effective DNA transfection in vivo [166]. PBAE-based NPs enabled high and preferential DNA delivery to HCC cells, sparing healthy hepatocytes [166]. However, the therapeutic efficacy of this approach was not tested. A novel polymer-nanoparticle-liquid ternary size-changing cationic nanodroplet (TNDs) was developed to deliver microRNA-122 (miR-122) for HCC treatment [62]. After treatment with PFP (Perfluoropentane)-TNDs/miR-122 combined with ultrasound irradiation, the miR-122 expression level was significantly increased in HCC cells and in human HCC xenografts, and inhibited tumor growth in mice [62]. In another study, antisense-miRNA-21 and gemcitabine (GEM) co-encapsulated PEGylated-PLGA polymer NPs were synthesized and formulated for in vitro therapeutic efficacy in human HCC (Hep3B and HepG2) cells [167]. This approach demonstrated an increase in vitro cell cycle arrest and apoptosis, but was not tested in vivo. Poly(beta-amino-ester) (PBAE) NPs were used to deliver a completely CpG-free plasmid harboring mutant herpes simplex virus type 1 sr39 thymidine kinase (sr39) DNA driven by AFP promoter to human HCC cells [168]. CpGf-AFP-sr39 NP treatment resulted in a 62% reduction in the size of orthotopic Hep3B xenografts [168]. Polymeric NPs incorporating AFP siRNA, along with angiogenesis inhibitor sunitinib, demonstrated a synergistic effect in decreasing cell viability of HCC cells in vitro [169]. Hyperbranched polyamidoamine was synthesized via a simple and economically one-pot reaction followed by decoration with lactobionic acid (LA-PAMAM) to selectively deliver and restore miR-218 expression in HCC cells [170]. LA-PAMAM/pmiR-218 treatment inhibited DEN/CCl_4_-induced HCC in vivo. Lactobionic acid-decorated PAMAM dendrimers delivering siRNA against Astrocyte-elevated gene-1/metadherin (AEG-1/MTDH) markedly inhibited the growth of orthotopic HCC xenografts alone and in combination with all-trans retinoic acid [171].

### 5.3. Safety and Immunogenicity Profiles

The use of viral and non-viral vectors in gene therapy to deliver genes for treating tumors is promising and has been widely tested in preclinical and clinical studies. However, there are still concerns regarding their safety, efficiency, and immunogenicity. Viral vectors are highly efficient at delivering genetic material into cells, but their viral origin also means they can trigger immune responses and carry other safety concerns, such as insertional mutagenesis. Different factors, including vector design, dose, and route of administration, contribute to the overall immunogenicity of these vectors. For AAVs, a high dose of vectors has shown strong immune responses, such as complement activation and liver toxicity, leading to death [172]. Although AAV vectors induce only marginal innate responses below the threshold of systemic symptoms, recent trials have shown that complement activation can result in serious adverse events [173]. Immune responses to other viral vectors have also been demonstrated [174]. Previous clinical or sub-clinical exposure to viruses, especially to AAV and AdV, can lead to development of pre-existing immunity, such as neutralizing antibody (NAB), which can significantly reduce the efficacy of virus-mediated gene delivery [174]. AdV serotype 5, the most commonly used AdV, uses the coxsackie and adenovirus receptor (CAR) to infect cells [175]. It has been demonstrated that CAR functions as a tumor suppressor and is significantly downregulated in HCC patients, thereby reducing the efficacy of AdV serotype 5-mediated gene therapy [175]. Chimeric AdV can potentially solve this problem. Replacing hexon hypervariable regions (HVRs) in the capsid of AdV serotype 1 and 5 with those of AdV serotype 35, along with alterations in the fiber region, not only improved infection efficiency but also enabled the virus to persist in the presence of NAB [176]. Chimera of AdV serotypes 5 and 3 have also been shown to avoid NAB and infect tumor cells efficiently [177]. Non-viral vectors are generally considered as safer with lower immunogenicity compared to viral vectors because they do not contain viral proteins. This reduces the risk of triggering an innate or adaptive immune response against the delivery vehicle. As such, non-viral vectors might be suitable for HCC patients with underlying autoimmune disease [16]. In a pre-clinical model of HCC, the feasibility and efficacy of local delivery of polymeric nanoparticles via intra-articular injection have been demonstrated [178], suggesting that non-viral vectors might be used in advanced HCC patients belonging to Child-Pugh Class B [168], which needs to be confirmed by performing clinical trials. While the vector itself is less immunogenic, the delivered genetic material (e.g., plasmid DNA, mRNA) can still elicit an immune response. One of the major drawbacks of nonviral vectors, compared to viral vectors, is that they often require higher doses or repeated administration, potentially leading to systemic exposure, toxicity, and off-target effects [179]. Factors like degradation of the genetic material, lack of integration into the host genome, or limited cell division might necessitate repeated administration of a non-viral vector. Systemic administration of non-viral vectors can still lead to their distribution to unintended tissues [180]. At higher concentrations, some non-viral components (e.g., certain cationic polymers like PEI) can exhibit cytotoxicity.

In conclusion, choosing a vector for gene therapy involves a careful balance between safety and efficacy. Viral vectors offer high transduction efficiency and often long-term expression, but they carry higher risks of immunogenicity, insertional mutagenesis, and the generation of Replication-Competent Virus (RCV). Non-viral vectors generally have better safety profiles with lower immunogenicity and no risk of insertional mutagenesis, though they are often limited by lower transduction efficiency and transient expression. Ongoing research focuses on engineering safer and more efficient vectors by modifying existing platforms, developing novel delivery systems, and combining strategies to mitigate immune responses and improve targeting specificity.

### 5.4. Targeted Delivery Strategies

Various strategies are employed to achieve selective delivery and expression of therapeutic genes in HCC cells. Here we provide some examples.

#### 5.4.1. Transcriptional Targeting: Tumor-Specific Promoters (TSPs)

TSPs are active in specific cancer cells. Although they are active in different tumors, they are less active or inactive in normal cells. Examples of TSPs investigated for HCC are the AFP gene promoter/enhancer, the GPC3 promoter, and the TERT promoter [61,181,182]. Novel promoters, such as BIRC5/surviving promoter, have shown comparable efficacy to strong ubiquitous promoters in driving suicide gene expression in HCC models [183].

#### 5.4.2. Ligand-Mediated Targeting

Ligand-mediated targeting involves modifying vectors to attach ligands to their surfaces that bind specifically to receptors overexpressed on the surface of HCC cells. One such example is ASGPR, which is normally expressed in hepatocytes but is particularly overexpressed in HCC. This differential distribution of ASGPRs was exploited in targeting HCC with several ligands, including galactose (GAL) and its derivatives, lactose, lactoferrin, and lactobionic acid (LA) [184]. Folic acid receptors are upregulated in HCC and, therefore, have been exploited in active targeting. In different studies, siRNAs were successfully delivered to HCC cells using folate-targeted selenium nanoparticles (SeNPs) [185,186].

## 6. Clinical Trials for Gene Therapy and Outcomes of HCC

Gene therapy holds significant promise for treating HCC, and numerous clinical trials have explored its potential using both viral and non-viral vectors. Table 1, Table 2 and Table 3 summarize the results of several completed and ongoing clinical trials that use adenoviruses, Chimeric Antigen Receptor T (CAR-T) cells, and JX-594. Ongoing trials are being conducted globally, with significant activity in China, North America, and Europe [31]. Most ongoing trials target patients with unresectable HCC, reflecting the urgent need for effective therapies in this population [31]. Notably, clinicaltrials.gov did not provide the genetic and/or molecular profiles of the HCC, highlighting a factor that should be considered for future standardization of trial reporting.

## 7. Challenges, Future Directions, and Perspectives

Despite the promise of using gene therapy in HCC, several challenges preclude its effective clinical use. One of these challenges is to ensure that therapeutic genes are delivered effectively and specifically to cancer cells while minimizing off-target effects. The immune system can recognize viral vectors and induce neutralization, which limits the duration and efficacy of gene therapy. HCC is highly heterogeneous, meaning that different cells within the same tumor can have varying genetic mutations, making it difficult to target all cancer cells with a single approach. Combination therapies are designed to address this issue by targeting multiple pathways. The focus for future research will be on (a) developing safer and more efficient, biocompatible and biodegradable gene delivery systems, (b) identifying optimal combinations of therapeutic genes and other agents, (c) integrating gene therapy with other treatment modalities (e.g., immunotherapy, targeted therapy), (d) personalizing combination gene therapy based on the specific genetic profile of a patient’s HCC, (e) addressing resistance mechanisms by understanding how HCC develops resistance to gene therapy, (f) developing strategies to overcome resistance, and (g) investigating emerging targets in HCC signaling pathways to explore more gene therapy for HCC prevention and treatment [187].

## 8. Conclusions

HCC is one of the leading causes of cancer-related deaths worldwide. Traditional treatments for HCC, such as surgical resection and systemic therapies, often yield limited efficacy and significant side effects, particularly in advanced stages. Gene therapy emerges as a potential alternative, targeting specific genetic alterations and employing advanced delivery systems to enhance therapeutic outcomes. Current gene therapy for HCC includes approaches such as CAR-T cells, LNP-siRNA, oncolytic viruses, CRISPR/Cas9 gene editing, and suicide gene therapy. It is showing promise in preclinical studies and with ongoing clinical trials exploring various vectors and strategies to enhance treatment efficacy. However, challenges remain in safety, immune response, delivery efficiency, and transgene expression. Future research is essential to clarify the efficacy and safety of these innovative gene treatments and to explore genetic targets to improve treatment outcomes and address limitations of existing therapies.

## Figures and Tables

**Figure 1 cancers-17-03608-f001:**
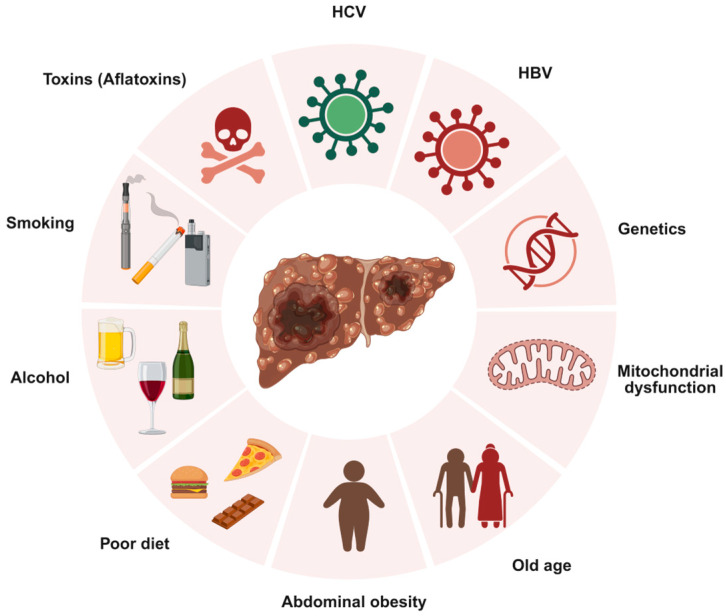
Overview of key risk factors associated with hepatocellular carcinoma (HCC). Created with BioRender.com. Production link: https://app.biorender.com/illustrations/68ec98dfb43a28ba87a0a942?slideId=1bcb511e-3401-48cf-8222-b2f76ea42d0d on 13 October 2025.

**Figure 2 cancers-17-03608-f002:**
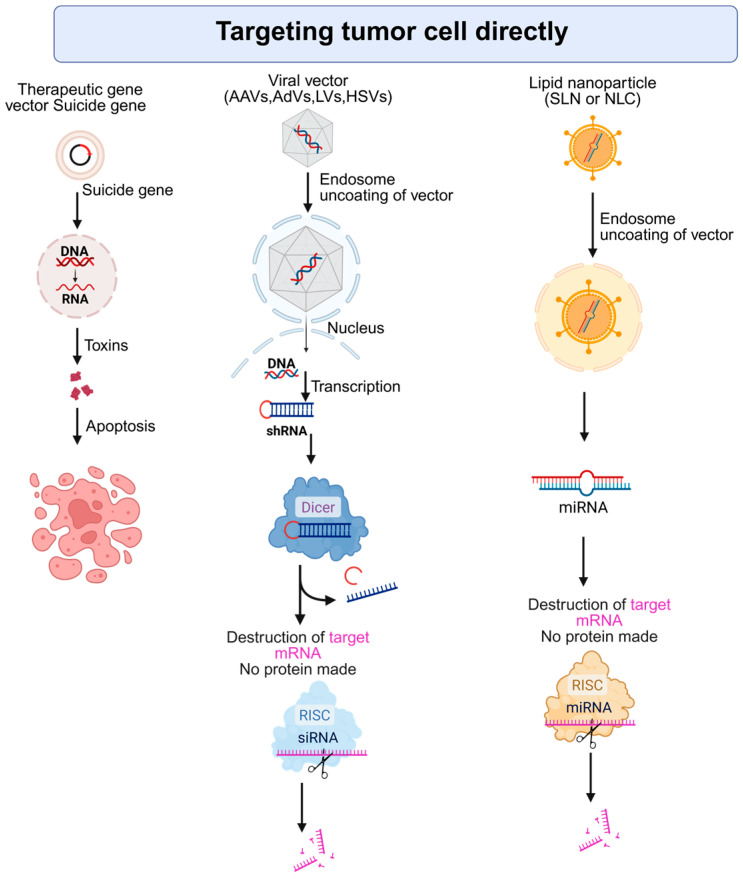
Strategies for direct targeting of tumor cells using gene and RNA-based therapies. AAV: Adeno-associated virus, AdV: Adenovirus, LV: Lentivirus, HSV: Herpes simplex virus, RISC: RNA-induced silencing complex, SLN: Solid Lipid Nanoparticles, NLC: Nanostructured Lipid Carriers. Created with BioRender.com. Production link: https://app.biorender.com/illustrations/68ed3a33968efac1ccb16c63?slideId=005ab124-32b1-4730-a0db-01f8d9e8f136 on 5 November 2025.

**Figure 3 cancers-17-03608-f003:**
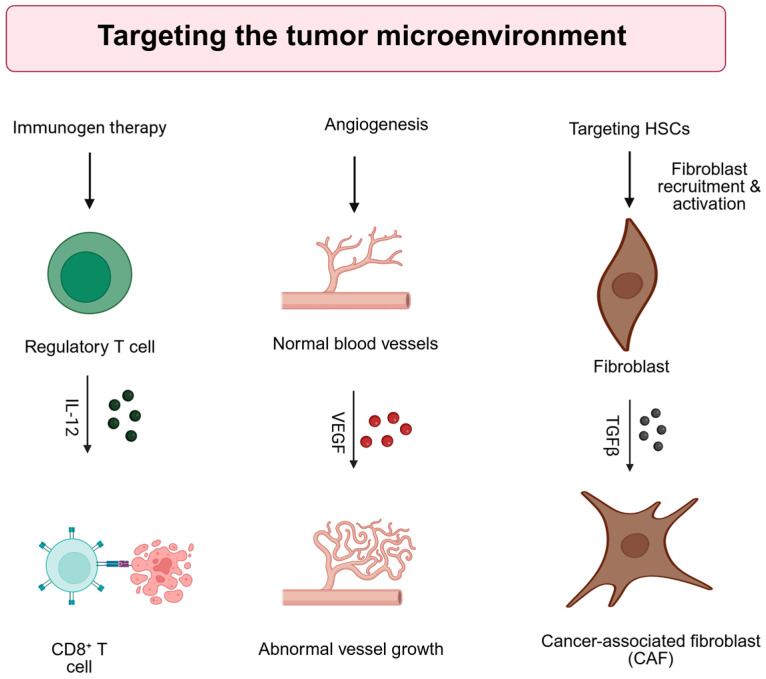
Strategies for direct targeting of the tumor microenvironment (TME). IL-12: interleukin-12, VEGF: Vascular endothelial growth factor, TGFb: transforming growth factor b, HSC: hepatic stellate cell. Created with BioRender.com. Production link: https://app.biorender.com/illustrations/68ee59ff04729264cecd5d90?slideId=005ab124-32b1-4730-a0db-01f8d9e8f136 on 14 October 2025.

**Table 1 cancers-17-03608-t001:** Data collected from clinicaltrials.gov as of 10 October 2025, showing national clinical trials (NCTs) that use Adenoviruses to treat HCC.

NCT Number	Study Status	Disease Conditions	Interventions	Phases	Enrollment	Locations
NCT00844623	Completed	HCC	Adenoviral vector containing thymidine kinase (TK99UN)	1	10	Spain
			Recombinant human adenovirus			
NCT05872841	Unknown	Primary HCC	rhAdV type 5 + TACE	2	38	China
NCT01869088	Unknown	HCC	rhAdV type 5	3	266	China
NCT05113290	Unknown	HCC	rhAdV type 5 + Sorafenib	4	66	China
NCT03790059	Unknown	HCC	H101 oncolytic virus	NA	160	China
NCT06685354	Not yet recruiting	HCC	rhAdV type 5	2	11	China
NCT00669136	Terminated	HCC, Hepatoma, Liver Cancer	AFP + GM-CSF Plasmid Prime and AFP Adenoviral Vector Boost	1	2	United States (USA)
NCT03780049	Unknown	HCC	H101|HAIC of FOLFOX| DRUG: DRUG: Placebos	3	304	China
NCT05675462	Recruiting	HCC	Human recombinant type 5 adenovirus (Oncorine) Tislelizumab+Lenvatinib	1	25	China
NCT02202564	Completed	Liver Cancer|HCC| Liver Transplantation	AdV-TK|ganciclovir	2	81	China
NCT00300521	Completed	HCC|Liver Transplantation	AdV-TK	2	40	China
NCT03313596	Unknown	HCC	AdV-Tk	3	180	China
NCT04612504	Unknown	HCC	Recombinant oncolytic adenovirus SynOV1.1	1	15	China
NCT02509169	Unknown	Advanced HCC	Transcatheter arterial embolization (TAE) plus P53 gene	2	60	China
NCT00003147	Terminated	Liver Cancer	Ad5CMV-p53 gene	1	30	USA
NCT02561546	Unknown	HCC|Diabetes	Recombinant adenovirus *p53* injection (rAdp53) Trans-catheter embolization	2	40	China
NCT00093548	Withdrawn	Liver Cancer	Alpha fetoprotein adenoviral vector vaccine	1, 2	0	USA

**Table 2 cancers-17-03608-t002:** Data collected from clinicaltrials.gov as of 10 October 2025, showing NCTs that use Chimeric Antigen Receptor T (CAR-T) cells to treat HCC.

NCT Number	Study Status	Disease Conditions	Interventions/Drugs	Phases	Enrollment	Locations
NCT05620706	Recruiting	Advanced HCC	GPC3 CAR-T cells	NA	20	China
NCT03884751	Completed	HCC	CAR-GPC3 T Cells	1	9	China
NCT03146234	Completed	HCC	CAR-GPC3 T cells	NA	7	China
NCT03980288	Completed	Advanced HCC	CAR-GPC3 T Cells	1	6	China
NCT04951141	Unknown	HCC	anti-GPC3 CAR-T cells	1	10	China
NCT02715362	Unknown	HCC	TAI-GPC3-CART cells	1, 2	30	China
NCT05103631	Recruiting	Liver Cancer	CATCH T cells	1	27	USA
NCT06198296	Not yet recruiting	HCC	21.15.GPC3-CAR T cells	1	21	USA
NCT03198546	Recruiting	HCC	GPC3 and/or TGFβ targeting CAR-T cells	1	30	China
NCT02905188	Completed	HCC	GLYCAR T cells|DRUG: Cytoxan|DRUG: Fludarabine	1	9	USA
NCT05926726	Recruiting	HCC	CAR-GPC3 T cells	NA	12	China
NCT03130712	Unknown	HCC	GPC3-CART cells	1, 2	10	China
NCT06144385	Recruiting	Liver Cancer|HCC	CAR-GPC3 T cells	1	20	China
NCT06641453	Not yet recruiting	HCC	GPC3-CART cells|DRUG: Fludarabine Phosphate for Injection|DRUG: Cyclophosphamide for Injection	1, 2	30	China
NCT05003895	Recruiting	HCC	Cyclophosphamide|CAR-T cell|Fludarabine	1	38	USA
NCT06461624	Recruiting	Advanced HCC	anti-GPC3 CAR-T	1	15	China
NCT03302403	Unknown	HCC	CAR-CD19 T cell|CAR-BCMA T cell|CAR-GPC3 T cell|CAR-CLD18 T cell|Fludarabine|Cyclophosphamide	NA	18	China
NCT02395250	Completed	Advanced HCC	anti-GPC3 CAR T	1	13	China
NCT02959151	Unknown	HCC	CAR-T cell	1, 2	20	China
NCT06968195	Not yet recruiting	Advanced HCC	Anti-GPC3-CAR Autologous TL	1	24	USA
NCT05120271	Recruiting	HCC	CAR-GPC3 T Cells	1, 2	110	USA, United Kingdom (UK)
NCT06560827	Recruiting	HCC	CT011 CAR-GPC3 T Cells	1	30	China
NCT05783570	Recruiting	Advanced HCC	EU307 CAR-T Cell	1	12	Korea Republic
NCT05652920	Recruiting	HCC	Ori-C101 (GPC3-directed chimeric antigen receptor modified T cells)	1, 2	105	China
NCT05155189	Recruiting	HCC	C-CAR031|DRUG: Lenvatinib|PD-1(L1) monoclonal antibody	1	44	China
NCT06891742	Recruiting	HCC	GPC3-targeted chimeric antigen receptor autologous T cell injection (OriC902)	1	44	China

**Table 3 cancers-17-03608-t003:** Data collected from clinicaltrials.gov as of 10 October 2025, showing NCTs that use JX-594 (Pexa-Vec) to treat HCC.

NCT Number	Study Status	Disease Conditions	Interventions/Drugs	Phases	Enrollment	Locations
NCT01171651	Completed	HCC	JX-594 followed by sorafenib	2	25	Korea Republic
NCT01636284	Completed	HCC	JX-594 recombinant vaccina GM-CSF	2	16	USA, Korea, Spain
NCT00554372	Completed	HCC	JX-594: Recombinant vaccinia virus (TK-deletion plus GM-CSF)	2	30	USA, Canada, Korea Republic,
NCT02562755	Completed	HCC	Pexastimogene Devacirepvec (Pexa Vec)|DRUG: Sorafenib	3	459	USA, UK
NCT01387555	Completed	HCC	JX-594 recombinant vaccina GM-CSF|OTHER: Best Supportive Care	2	129	USA, Canada, France, Germany, Korea, Taiwan
NCT00629759	Completed	Neoplasms, Liver	JX-594: Recombinant vaccinia virus (TK-deletion plus GM-CSF)	1	14	Korea Republic

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
