# Peer review of "Gene Therapy Strategies for Hepatocellular Carcinoma (HCC): Current Landscape and Future Directions"

_cancers, 2025, doi:10.3390/cancers17223608_

Round 1

Reviewer 1 Report

Comments and Suggestions for Authors

In this review by Ali Gawi Ermi et al., the authors summarize recent literature describing gene therapy strategies to manage hepatocellular carcinoma (HCC), increase survival and improve HCC patient’s quality of life.

They started by introducing HCC, its epidemiology, etiology, and existing clinical challenges for its treatment, then introducing the rationale of using gene therapy approaches for HCC treatment. They describe strategies to target directly tumor cells (suicide gene therapy with or without combination therapies, restoration of immunosuppressors, oncolytic virotherapy, gene editing approaches), followed by strategies targeting the tumor microenvironment (immunogen therapy, anti-angiogenic gene therapy, targeting of hepatic stellate cells); then they introduce Chimeric Antigen Receptor (CAR)-T cells, Vector Design and Delivery Systems describing both viral and non-viral vector approaches, and then finally describe the Clinical Trials for Gene Therapy and Outcomes of HCC.

Overall, this review offers a comprehensive overview of direct and indirect gene therapy approaches aimed at treating HCC. It is well written and informative.

However, the authors should mention problem related to the use of viral vectors, such as possible lower efficiency of transduction of transformed/tumor cells and the difficulties linked to the use of particular viral vectors, such as pre-existing immunity to particular AAV and Ad serotypes and problems associated with multiple infusions for further treatments.

Author Response

We thank the reviewers for their astute comments and suggestions and finding our manuscript as an appropriate review of the field. We addressed the reviewers’ comments and concerns editing the text and modifying Figure 2. We believe the revised manuscript is significantly improved. A point-by-point response is provided below:

Reviewer#1

Comment 1. The authors should mention problem related to the use of viral vectors, such as possible lower efficiency of transduction of transformed/tumor cells and the difficulties linked to the use of particular viral vectors, such as pre-existing immunity to particular AAV and Ad serotypes and problems associated with multiple infusions for further treatments.

Answer: We thank the reviewer for this suggestion. These problems have been described in Section 5.3 of the revised manuscript, lines 798-809.

Reviewer 2 Report

Comments and Suggestions for Authors

Cancers

Comments on the review: Gene Therapy strategies for hepatocellular carcinoma (HCC): Current landscape and future directions by Ali Gawi Ermi et al.

Many compliments to the authors , because they have written a review on a hotspot of both cancer field and of liver tumors. In particular they have provided a current landscape and the future directions of the gene therapy strategies for HCC. Iappreciated very much the authors who have outlined ( in some paragraphs of the text) the necessity to know the genetic and molecular profiles of the HCC tumor before defining the therapeutic treatment, mainly in advanced HCC cases that are the most HCCs. In the text and in the table regarding the National Clinical Trials (conducted mainly in USA and China), I did not understand if these trials were/are conducted after knowing the genetic and/or molecular profiles of the HCC . I ask the authors to clarify this aspect and to outline even more the utility for clinicians to ask the genetic and molecular profiles of the HCC cancer before defining the therapeutic management for each patient. For other solid cancers (breast cancer, lung cancer..) and for blood malignancies (different types of leukemia) it is a clinical routine to know the gene mutations of the tumor for each cancer patient.  For HCC the situation is more complex, but it is necessary to speed up the molecular knowledge of the cancer mass.

Further I have other small requests.

Simple summary. I ask to include ( together with liver transplantation) also the HCC resection as therapeutic option for early HCC.

Figure 2 is not enough clear. I ask the authors to modify the scheme of viral vectors for transducing siRNA.

Paragraph 5.1 The authors report ...(like shRNA) to turn off ... It is necessary to write short hairpin RNA and use the abbreviation shRNA at paragraph 5.13.

Author Response

Reviewer#2

Comment 1: In the text and in the table regarding the National Clinical Trials (conducted mainly in USA and China), I did not understand if these trials were/are conducted after knowing the genetic and/or molecular profiles of the HCC. I ask the authors to clarify this aspect and to outline even more the utility for clinicians to ask the genetic and molecular profiles of the HCC cancer before defining the therapeutic management for each patient.

Answer: We greatly appreciate the reviewers’ comments. Unfortunately, in ClinicalTrials.gov, for HCC clinical trials we did not find any information about analyzing genetic and/or molecular profiles of the patients. This information is included in Section 6, lines 863-865.

Comment 2: Simple summary. I ask to include (together with liver transplantation) also the HCC resection as therapeutic option for early HCC.

Answer: We modified the Simple Summary as suggested by the reviewer.

Comment 3: Figure 2 is not enough clear. I ask the authors to modify the scheme of viral vectors for transducing siRNA.

Answer: We apologize for the lack of clarity. We now provide better image.

Comment 4: Paragraph 5.1 The authors report …(like shRNA) to turn off … It is necessary to write short hairpin RNA and use the abbreviation shRNA at paragraph 5.13.

Answer: We thank the reviewer for pointing out the mistake which has been corrected in the revised manuscript.

Reviewer 3 Report

Comments and Suggestions for Authors

This paper is carefully structured. However, the gene therapy described herein is likely to be limited to cases of advanced hepatocellular carcinoma classified as Child-Pugh A, for which procedures such as liver resection, liver transplantation, radiofrequency ablation (RFA), and transarterial chemoembolisation (TACE) are not feasible. Immune checkpoint inhibitors and molecularly targeted drugs are currently used as primary treatments for advanced hepatocellular carcinoma. Molecularly targeted drugs are used when immune checkpoint inhibitors cannot be used, for example, in cases of autoimmune disease. So, could gene therapy be a realistic treatment for a weakened liver? Is it feasible for livers affected by autoimmune diseases? Do the authors consider gene therapy to be possible for Child-Pugh Class B patients? What side effects might be particularly pronounced in a weakened liver? Since the liver is almost entirely composed of hepatocytes, it has a high affinity for adenovirus. Why does drug efficacy vary despite the liver being largely homogeneous? Please state the authors' views.

Author Response

Reviewer#3

Comments: Could gene therapy be a realistic treatment for a weakened liver? Is it feasible for livers affected by autoimmune diseases? Do the authors consider gene therapy to be possible for Child-Pugh Class B patients? What side effects might be particularly pronounced in a weakened liver? Since the liver is almost entirely composed of hepatocytes, it has a high affinity for adenovirus. Why does drug efficacy vary despite the liver being largely homogeneous? Please state the authors' views.

Answer: We highly appreciate these comments and questions. We provide our views at the end of the second paragraph of Section 1.2 (lines 114-116) and in Section 5.3 (lines 798-809; 812-817).